# Total Synthesis of Lineaflavones A, C, D, and Analogues

**DOI:** 10.3390/molecules28052373

**Published:** 2023-03-04

**Authors:** Rui Wang, Yu Fu, Ran Ma, Hongzhen Jin, Wei Zhao

**Affiliations:** 1State Key Laboratory of Medicinal Chemical Biology, College of Pharmacy, Key Laboratory of Molecular Drug Research and KLMDASR of Tianjin, Nankai University, Tongyan Road, Haihe Education Park, Tianjin 300350, China; 2Tianjin International Joint Academy of Biomedicine, Tianjin 300457, China

**Keywords:** total synthesis, flavonoids, natural products, schenck ene reaction

## Abstract

The first total synthesis of lineaflavones A, C, D, and their analogues has been accomplished. The key synthetic steps include aldol/oxa-Michael/dehydration sequence reactions to assemble the tricyclic core, Claisen rearrangement and Schenck ene reaction to construct the key intermediate, and selective substitution or elimination of tertiary allylic alcohol to obtain natural compounds. In addition, we also explored five new routes to synthesize fifty-three natural product analogues, which can contribute to a systematic structure–activity relationship during biological evaluation.

## 1. Introduction

Flavonoids are a class of compounds that are produced by plants as secondary metabolites and exhibit important functions in reproduction; these compounds can be classified as flavanones, flavanols, isoflavones, flavones, and flavonols [1,2]. Modern pharmacological research has revealed that flavonoids have a wide range of bioactivities, such as antitumor, antioxidant, anti-inflammatory, and antivirus [1,3,4]. Flavonoids can affect the behavior of cellular systems by modulating the activity of enzymes, exerting beneficial effects on the organism [5]. A large variety of flavonoids have been isolated and characterized [6,7], but lead compounds with new biological activities remain to be discovered. The synthesis and identification of bioactive derivatives are crucial for intensive studies to discover lead compounds with novel structures.

Lineaflavones A, C, and D are natural flavonoids isolated from the aerial parts of *Tephrosialinearis* by Spiteller and co-workers in 2020 (Figure 1) [8]. The structures of the compounds were elucidated on the basis of their NMR and HRMS^n^ data. The anti-inflammatory effects of the isolated compounds were evaluated by measuring the levels of IL-6 and TNF-α and the tested compounds inhibited the production of IL-6 and TNF-α. Further study of these natural products and their analogues may provide new guidance for drug discovery. Due to their intriguing biological activity and unique structure; herein, the first total synthesis of lineaflavone A, lineaflavone C, and lineaflavone D was described. In addition, we also synthesized fifty-three flavonoid derivatives. Newly synthesized derivatives could be suitable for evaluating anti-inflammatory activity.

## 2. Results and Discussion

The isolated compounds with a 2″,2″-dimethylpyran ring and a linear side chain contain a C_6_-C_3_-C_6_ skeleton structure [9]. The protocol used to synthesize target molecules is depicted in the retrosynthetic analysis in Figure 1. Lineaflavones A and D could be synthesized from intermediate **4** through substitution or elimination. Construction of the key intermediate **4** by photooxygenation of prenylphenol followed by a reduction formed the structure of 3-hydroxy-3-methylbut-enyl based on the Schenck ene reaction and compound **5** as the reaction substrate [10]. Compound **5** could be derived from 5,7-dihydroxy-2-phenyl-4H-chromen-4-one (**6**), 3,3-dimethylallyl bromide (**7**), 3-chloro-3-methylbut-1-yne (**8**) and iodomethane (**9**) by the sequential chemical reactions. Compound **6** could be synthesized from the commercially available materials 1-(2,4,6-trihydroxyphenyl)ethan-1-one (**10**) and benzaldehyde (**11**) through aldol/oxa-Michael/dehydration sequence reactions [11,12].

The coupling of compounds **10** and **11** affords compound **6** via the aldol reaction, I_2_-catalyzed oxa-Michael addition reaction, and dehydration reaction, which may suffer from poor functional group tolerance to exposed hydroxyl groups and thus lead to the production of some other unknown side-products [12,13]. Accordingly, the free hydroxyl groups of substance **10** were protected using the protecting groups with satisfactory yields. Our initial strategy was to use methoxymethyl (MOM) to protect the C2′, C4′-hydroxy groups of **10** selectively in 92% yield, and the MOM-protected compound **10** was reacted with benzaldehyde **11** to afford the intermediate (78% yield), which was further reacted to obtain **6** by intramolecular addition reaction, but we only detected the yield of intermediate **6** from 10% to 18% by screening different temperatures and solvents. The possible reason for the low yield comes from the fact that the condition of the I_2_-catalyzed oxa-Michael addition reaction necessitates elevated temperatures, which can result in the removal of the MOM group. Therefore, we selected methyl with higher chemical stability as the protecting group, and compound **10** was converted to intermediate **14** by the sequential chemical reactions, resulting in a 68% overall yield (Figure 2). The deprotection of methyl groups of **14** was smoothly accomplished using HBr in AcOH to obtain compound **6** (78% yield) [14].

Chemoselective propargylation of the C7-hydroxyl of compound **6** provided **15** in 70% yield, which underwent an aromatic Claisen rearrangement to furnish cyclization products **16** (52% yield) and **17** (41% yield) under high-temperature conditions of 250 °C [15,16]. Then the cyclization product **17** was prenylated with 3,3-dimethylallyl bromide **7** to afford **18** (80% yield) (Figure 2). Compound **18** was subjected to the Claisen rearrangement reaction to establish the structure of **19,** and we did not detect any products using montmorillonite K10 or Bi(OTf)_3_·4H_2_O as the catalyst at the beginning [17]. Finally, we selected Eu(fod)_3_ as the catalyst in refluxing chloroform to obtain the intermediate **19** (76% yield). Methylation of the free hydroxy group of **19** with iodomethane using NaH gave the corresponding compound **20** in an 82% yield. Compound **20** was converted to **21** via photooxygenation of prenylphenol followed by reduction at room temperature, which can yield tertiary allylic alcohol and secondary allylic alcohol based on the Schenck ene reaction [10], therefore we also obtained the byproduct **22**. After a systematic investigation of catalysts and solvents, we determined to use Rose Bengal as a photosensitizer and MeOH as a solvent to give **21** and **22** in a 73% overall yield [18]. We turned to perform the Schenck ene reaction of **19** by using Rose Bengal, which only gave secondary allylic alcohol **23** as the main product (54% yield). An oxygen atom was introduced into compound **20** by alkene epoxidation with 3-chloroperoxybenzoic acid to produce compound **24** in 87% yield [19].

Having prepared tertiary allylic alcohol **21** successfully, we sought to construct the natural flavonoids **1** and **3**. We first attempted to convert **21** to **1** by nucleophilic substitution reaction. We used iodomethane and dimethyl sulfate as methylation agents. Even though various alkalis, including K_2_CO_3_, KOH, NaOH, and NaH, of this reaction were screened, we failed to detect the transformation of **21** into the desired compound **1**. One possible reason is that the tertiary alcohol has a strong steric hindrance effect. Given the fact that the tertiary alcohol is a typical substrate for dehydration, we then tried to use inorganic acid as a catalyst to achieve an elimination or substitution reaction. We only obtained the natural compound **3** with a 60% yield under the conditions of H_2_SO_4_ and MeOH in a 1:10 ratio (Table 1, entry 1). Changing the acid to HCl improved the yield (78% yield); however, we still detected compound **3** as the sole product (entry 2). After the systematic study of the reaction conditions, we found that decreasing the concentration of acid markedly improved the ratio of **1** (entries 3 and 4). When the concentration of HCl was further reduced, the proportion of **1** decreased, and the overall yield was also diminished (entries 5–7). The tertiary alcohol underwent transformation to give **1** as the major product by using the condition of entry 4. Based on the above research, we successfully synthesized the natural products lineaflavone A (**1**) (1.67% overall yield) and lineaflavone D (**3**) (2.50% overall yield). The spectroscopic data for synthetic **1** and **3** were well matched with those reported in the literature (Appendix A) [8].

The synthesis of natural product **2** from intermediate **19** is summarized in Figure 3. Initially, we attempted to remove the methyl group of **21**, followed by elimination, leading to compound **2**. To liberate the hydroxyl group of compound **21**, numerous attempts were made using different methods, including strong acids (HI and HBr), Lewis acids (AlCl_3_, BBr_3_, and BCl_3_), and inorganic base (NaNH_2_) [20,21]. None of these approaches provided the desired product. Thus, we selected the acetyl group to protect the C5-hydroxy group of **19.** Acetylation of intermediate **19** provided Ac-protected compound **25** (88% yield) and the Schenck ene reaction of the latter with the photosensitizer Rose Bengal yielded secondary allylic alcohol **26** (36% yield) and tertiary allylic alcohol **27** (41% yield), whose conversion to **28** was achieved by selective elimination (76% yield). Deacetylation of **28** was smoothly accomplished with LiOH in tetrahydrofuran from 0 °C to 60 °C to afford compound **2** in a 97% yield [22]. The spectroscopic data for synthetic **2** were consistent with those reported for the natural product (Appendix A) (3.51 % overall yield) [8].

The byproduct **22** has a unique hydroxyisoprenyl group, and flavonoids containing this structure usually exhibit biological activities according to the literature [23,24,25,26]. Thus, we synthesized a series of novel derivatives. Esterification of the hydroxyl group of **22** with various acyl chlorides by using 4-Dimethylaminopyridine (DMAP) gave the corresponding ester compounds **29a–29i** (yield from 65% to 83%) (Figure 4A). Beyond that, we introduced different side chains at the 8-position of compound **17** to increase the structural diversity. Compound **30** was prepared from **17** by treatment with allyl bromide in the presence of NaH in DMF at room temperature. The resultant **30** underwent rearrangement reactions to give **31** in a 51% yield in two steps. Compound **31** was coupled with commercially available reagents under typical alkene metathesis reaction conditions to give **32a–32c** (yield from 75% to 87%) (Figure 4B) [27].

Compound **15** was subjected to an aromatic Claisen rearrangement to give the byproduct **16**. After rationally screening different temperatures of the reaction, we found that the yield was enhanced from 52% to 95% at 120 °C, and compound **16** was the only product. Compound **16** was prenylated with 3,3-dimethylallyl bromide **7** to afford **33** (83% yield), which was further reacted in the presence of montmorillonite K10 to give the desired rearrangement product **34** (65% yield). Compounds **35a** and **35b** were synthesized from the methylation or esterification reaction of compound **34. The** Schenck ene reaction was carried out to obtain **36a**, **36c**, **37a,** and **37b**. Next, the secondary allylic alcohol **36c** was esterified and generated **40a–40e** via a substitution reaction with various acyl chlorides. Compounds **38** and **41** were prepared through an elimination or substitution reaction using inorganic acid as a catalyst. Removal of the Ac protecting group from **36a** and **38** with LiOH led to the final products **36b** and **39** (Figure 5A)**.** Compound **33** was also converted to **42** in the presence of Eu(fod)_3_ as a catalyst by the [3, 3] sigmatropic rearrangement reaction, and we synthesized a new series of analogues **43a**–**43f** by esterification or etherification of the free hydroxy group (yield from 65% to 89%) (Figure 5B) [28].

We also designed and synthesized a series of chalcone derivatives with new chemical structures. Compound **44** was selectively substituted with 3-chloro-3-methylbut-1-yne in DMF to produce **45** (78% yield), which was converted to **46** via an intramolecular cyclization (93% yield). Subsequent protection of the 5-OH of **46** with NaH and 3,3-dimethylallyl bromide afforded **47** in an 86% yield. Compound **47** was subjected to Claisen rearrangement, methylation, and aldol reactions to establish the structures of **50a**–**50c** (yield from 58% to 75%, three steps). Oxidation of **50a**–**50c** with tetraphenylporphyin (Tpp) and subsequent treatment of the resulting peroxide with PPh_3_ in one-pot afforded the corresponding compounds **51a**–**51c** and **52a**–**52c** (yield from 35% to 49%). Interestingly, the tertiary allylic alcohol underwent a further transformation to give the diene under Schenck ene reaction conditions (Figure 6). Beyond that, we removed the 2″,2″-dimethylpyran ring structure from compound **49** and synthesized the chalcone derivative **58** from the commercially available material **44** through the sequential chemical reactions in a 38% overall yield (Figure 7).

## 3. Experimental Section

### 3.1. General Information

Unless otherwise stated, all reactions were carried out under an argon atmosphere with dry solvents under anhydrous conditions. Dimethylformamide (DMF) and dichloromethane (CH_2_Cl_2_) were distilled from calcium hydride and stored under argon. All other reagents were purchased at the highest commercial quality and used without further purification. Flash chromatography was performed using 200–400 mesh silica gel. Analytical thin-layer chromatography (TLC) was performed on Merck silica gel 60 F254 aluminum sheets. TLC was visualized by one of the following methods: use of UV light (254 nm), exposure to iodine vapor, or treatment of acidic anisaldehyde.

NMR spectra were recorded on Bruker 400 MHz instruments and calibrated using residual solvent as an internal reference (^1^H NMR: CDCl_3_ = 7.26, DMSO-*d*_6_ = 2.50, Acetone-*d*_6_ = 2.05 and ^13^C NMR: CDCl_3_ = 77.16, DMSO-*d*_6_ = 39.52, Acetone-*d*_6_ = 29.84, 206.26). The coupling constant was reported in Hertz units (Hz). The following abbreviations were used to explain the multiplicities: s = singlet, d = doublet, t = triplet, q = quartet, and m = multiplet. High-resolution mass spectra (HRMS) were obtained on an IonSpec QFT mass spectrometer with ESI ionization.

### 3.2. Materials

All solvents and commercially available chemicals were used as received without further purification unless otherwise stated.

### 3.3. Procedure for the Synthesis of Lineaflavones A, C, D

To a solution of 1-(2,4,6-trihydroxyphenyl)ethan-1-one (2.00 g, 11.89 mmol) in anhydrous acetone (50 mL) was added K_2_CO_3_ (3.62 g, 26.17 mmol) and dimethyl sulfate (2.32 mL, 24.38 mmol). The reaction mixture was stirred at 60 °C for 4 h. The resulting mixture was cooled to room temperature, filtered, and washed with acetone. The filtrate was extracted with EtOAc (250 mL). The organic layer was washed three times with brine (100 mL × 3), dried over anhydrous Na_2_SO_4_, filtered, and concentrated in vacuo. The crude product was purified by flash chromatography (hexane:EtOAc = 20:1) to afford **12** (2.10 g, 90% yield) as a white solid.

^1^H NMR (400 MHz, Acetone-*d*_6_) δ 13.97 (s, 1H), 6.07 (d, *J* = 2.7 Hz, 1H), 6.03 (d, *J* = 2.7 Hz, 1H), 3.92 (s, 3H), 3.85 (s, 3H), and 2.56 (s, 3H).

^13^C NMR (100 MHz, Acetone-*d*_6_) δ 203.88, 168.43, 167.41, 164.21, 106.52, 94.44, 91.40, 56.08, and 33.05.

To a solution of compound **12** (5.00 g, 25.50 mmol) and benzaldehyde (9.02 mL, 76.50 mmol) in EtOH (250 mL) was added NaOH (2.01 g, 50.10 mmol) at 0 °C. After stirring for 0.5 h, the resulting solution was stirred at 50 °C for 24 h before the addition of water and EtOAc. The aqueous phase was extracted three times with EtOAc (200 mL × 3), and the organic layers were successively washed three times with brine (100 mL × 3), dried over anhydrous Na_2_SO_4_, filtered, and concentrated in vacuo. The crude product was purified by flash chromatography (hexane:EtOAc = 15:1) to afford **13** (6.45 g, 89% yield) as a yellow solid.

^1^H NMR (400 MHz, Acetone-*d*_6_) δ 14.23 (s, 1H), 8.04 (d, *J* = 15.6 Hz, 1H), 7.82–7.73 (m, 3H), 7.54–7.39 (m, 3H), 6.14 (d, *J* = 2.1 Hz, 1H), 6.12 (s, 1H), 4.02 (s, 3H), and 3.89 (s, 3H).

^13^C NMR (100 MHz, Acetone-*d*_6_) δ 192.53, 168.29, 166.74, 162.91, 142.13, 135.51, 130.22, 129.00, 128.42, 127.46, 105.95, 93.79, 91.00, 55.68, and 55.24.

Compound **13** (0.3 g, 1.06 mmol) and iodine (26.90 mg, 0.11 mmol) were stirred in DMSO (25 mL) at 170 °C for 3 h. Then, the mixture was poured into an 80 mL solution of 10% Na_2_S_2_O_3_ and stirred. The precipitate was collected by filtration and washed with hexane. The crude product was recrystallized from ethanol and water (1:1) to yield the pure product **14** (0.25 g, 85%). Compound **14** was stirred in hydrobromic acid (33 wt.% solution in acetic acid) at 120 °C for 24 h. H_2_O and EtOAc were added to the reaction mixture, then the aqueous phase was extracted three times with EtOAc (150 mL × 3), and the organic layers were washed three times with brine (80 mL × 3), dried over anhydrous Na_2_SO_4_, filtered, and concentrated in vacuo. The crude product was purified by flash chromatography (hexane:EtOAc = 3:1) to afford **6** (0.21 g, 78% yield) as a yellow solid.

^1^H NMR (400 MHz, DMSO-*d*_6_) δ 12.82 (s, 1H), 10.90 (s, 1H), 8.18–7.92 (m, 2H), 7.72–7.47 (m, 3H), 6.93 (s, 1H), 6.51 (s, 1H), and 6.22 (s, 1H).

^13^C NMR (100 MHz, DMSO-*d*_6_) δ 182.32, 164.91, 163.61, 161.95, 157.92, 132.44, 131.18, 129.52, 126.81, 105.63, 104.44, 99.49, and 94.58.

To a suspension of compound **6** (0.50 g, 1.97 mmol) in DMF (50 mL) was added K_2_CO_3_ (0.54 mg, 3.93 mmol), KI (0.49 mg, 2.95 mmol), CuI (18.75 mg, 0.09 mmol), and 3-chloro-3-methylbut-1-yne (0.42 mL, 3.74 mmol) at room temperature for 2 h. The resulting mixture was a light red solution. The reaction mixture was quenched by saturated aqueous NH_4_Cl (20 mL). The layers were separated, and the aqueous layer was extracted three times with EtOAc (50 mL × 3). The combined organic layers were washed with brine (30 mL × 3), dried over anhydrous Na_2_SO_4_, filtered, and concentrated in vacuo. The crude product was purified by flash chromatography (hexane:EtOAc = 10:1) to afford **15** (0.44 g, 70% yield) as a yellow solid.

^1^H NMR (400 MHz, Acetone-*d*_6_) δ 12.80 (s, 1H), 8.11–8.05 (m, 2H), 7.66–7.53 (m, 3H), 7.00 (d, *J* = 2.2 Hz, 1H), 6.83 (s, 1H), 6.62 (d, *J* = 2.2 Hz, 1H), 3.39 (s, 1H), and 1.75 (s, 6H).

^13^C NMR (100 MHz, Acetone-*d*_6_) δ 182.50, 164.21, 162.12, 161.64, 157.22, 131.98, 131.28, 129.15, 126.47, 105.95, 105.49, 102.29, 97.47, 84.68, 76.39, 72.88, and 29.36.

A solution of **15** (280.40 mg, 0.88 mmol) in diethylaniline (20 mL) was stirred at 250 °C for 1 h. The resulting mixture was cooled to room temperature, and EtOAc was added to the reaction mixture. The organic layers were extracted three times with 1N HCl solution (100 mL × 3), and the organic layers were washed with brine (30 mL × 3), dried over anhydrous Na_2_SO_4_, filtered, and concentrated in vacuo. The crude product was purified by flash chromatography (hexane:EtOAc = 20:1) to afford **17** (114.96 mg, 41% yield) and **16** (145.81 mg, 52% yield) as a yellow solid.

**17:** ^1^H NMR (400 MHz, Chloroform-*d*) δ 13.03 (s, 1H), 7.98–7.78 (m, 2H), 7.58–7.45 (m, 3H), 6.72 (d, *J* = 9.6 Hz, 1H), 6.63 (s, 1H), 6.42 (s, 1H), 5.62 (d, *J* = 9.6 Hz, 1H), and 1.48 (s, 6H).

^13^C NMR (100 MHz, Chloroform-*d*) δ 182.56, 163.75, 159.61, 157.15, 156.45, 131.78, 131.37, 129.09, 128.18, 126.26, 115.50, 105.73, 105.67, 105.60, 95.12, 78.05, and 28.32.

**16:**^1^H NMR (400 MHz, Chloroform-*d*) δ 12.79 (s, 1H), 8.10–7.75 (m, 2H), 7.62–7.48 (m, 3H), 6.79 (dd, *J* = 9.9, 1.5 Hz, 1H), 6.64 (s, 1H), 6.27 (s, 1H), 5.62 (dd, *J* = 9.9, 1.5 Hz, 1H), and 1.49 (s, 6H).

^13^C NMR (100 MHz, Chloroform-*d*) δ 182.69, 163.49, 161.79, 159.68, 151.98, 131.87, 131.45, 129.17, 127.60, 126.21, 114.83, 105.87, 105.52, 101.39, 100.40, 78.10, and 28.22.

To a solution of **17** (1.70 g, 5.31 mmol) in anhydrous DMF (50 mL) was added NaH (0.64 g, 15.92 mmol) and 3,3-dimethylallyl bromide (1.14 mL, 10.61 mmol). The reaction mixture was stirred at room temperature for 6 h. The reaction mixture was quenched with brine (50 mL). The aqueous layer was extracted three times with EtOAc (50 mL × 3). The combined organic layers were washed three times with brine (30 mL × 3), dried over anhydrous Na_2_SO_4_, filtered, and concentrated in vacuo. The crude product was purified by flash chromatography (hexane:EtOAc = 8:1) to afford **18** (1.65 g, 80% yield) as a yellow solid.

^1^H NMR (400 MHz, Acetone-*d*_6_) δ 8.10–7.95 (m, 2H), 7.69–7.46 (m, 3H), 6.80 (s, 1H), 6.73 (d, *J* = 10.3 Hz, 1H), 6.66 (s, 1H), 5.86 (d, *J* = 10.3 Hz, 1H), 5.59 (m, 1H), 4.60 (d, *J* = 7.3 Hz, 2H), 1.75 (s, 3H), 1.67 (s, 3H), and 1.47 (s, 6H).

^13^C NMR (100 MHz, Acetone-*d*_6_) δ 175.74, 160.46, 158.67, 157.93, 154.01, 137.63, 131.58, 131.30, 130.59, 129.05, 125.99, 120.70, 116.47, 113.69, 112.83, 108.04, 100.54, 77.53, 71.47, 27.48, 25.09, and 17.19.

To a solution of **18** (0.10 g, 0.26 mmol) in anhydrous chloroform (20 mL) was added Eu(fod)_3_ (14 mg, 0.01 mmol). The resulting orange solution was stirred at 60 °C for 8 h and then the solvent was removed under reduced pressure. The crude product was purified by flash chromatography (hexane:EtOAc = 15:1) to afford **19** (76 mg, 76% yield) as a white solid.

^1^H NMR (400 MHz, Acetone-*d*_6_) δ 13.21 (s, 1H), 8.16–7.95 (m, 2H), 7.72–7.50 (m, 3H), 6.79 (s, 1H), 6.66 (d, *J* = 10.0 Hz, 1H), 5.77 (d, *J* = 10.0 Hz, 1H), 5.24 (m, 1H), 3.53 (d, *J* = 7.1 Hz, 2H), 1.84 (s, 3H), 1.66 (s, 3H), and 1.48 (s, 6H).

^13^C NMR (100 MHz, Acetone-*d*_6_) δ 182.78, 163.74, 156.80, 154.47, 154.42, 131.88, 131.58, 131.28, 129.19, 128.40, 126.38, 122.30, 115.23, 107.59, 105.23, 105.20, 105.02, 77.91, 27.46, 24.99, 21.32, and 17.37.

IR (thin film): 2361, 2338, 1651, 1458, 1344, 1308, 873, 722, and 652 cm^−1^

m.p.: 485.8 °C

To a solution of **19** (0.60 g, 1.54 mmol) in anhydrous DMF (50 mL) was added NaH (0.86 g, 6.16 mmol) and CH_3_I (0.19 mL, 3.08 mmol). The reaction mixture was stirred at room temperature for 3 h. The reaction mixture was quenched with brine (50 mL). The aqueous layer was extracted three times with EtOAc (250 mL × 3). The combined organic layers were washed with brine (100 mL × 3), dried over anhydrous Na_2_SO_4_, filtered, and concentrated in vacuo. The crude product was purified by flash chromatography (hexane:EtOAc = 10:1) to afford **20** (0.51 g, 82% yield) as a yellow solid.

^1^H NMR (400 MHz, Acetone-*d*_6_) δ 8.14–7.92 (m, 2H), 7.70–7.46 (m, 3H), 6.74 (d, *J* = 10.1 Hz, 1H), 6.66 (s, 1H), 5.89 (d, *J* = 10.1 Hz, 1H), 5.32–5.22 (m,1H), 3.84 (s, 3H), 3.61 (d, *J* = 7.0 Hz, 2H), 1.85 (s, 3H), 1.67 (s, 3H), and 1.49 (s, 6H).

^13^C NMR (100 MHz, Acetone-*d*_6_) δ 175.94, 160.49, 156.05, 155.11, 153.14, 131.94, 131.65, 131.28, 130.91, 129.11, 126.02, 121.98, 116.04, 113.43, 112.75, 112.45, 107.94, 77.54, 61.91, 27.44, 25.01, 21.88, and 17.45.

IR (thin film): 3068, 2921, 1699, 1654, 1242, 1165, 1121, 1099, 1027, 872, 689, and 650 cm^−1^

m.p.: 420.2 °C

Dried air was continuously bubbled through a MeOH (60 mL) solution of **20** (0.20 g, 0.49 mmol) and Rose Bengal (25.20 mg, 0.03 mmol) as the photosensitizer. A 500 W halogen lamp was used as the light source. The reaction mixture was irradiated and stirred at room temperature for 10 h. The crude residue was directly used without further purification. Triphenylphosphine (0.19 g, 0.74 mmol) was added, and the solution was stirred at room temperature for 16 h before being concentrated in vacuo. The crude residue was purified by flash chromatography (hexane:EtOAc = 4:1) to afford **21** (72.78 mg, 35%) and **22** (78.83 mg, 38%) as a white solid.

**21:** ^1^H NMR (400 MHz, Acetone-*d*_6_) δ 8.10–8.04 (m, 2H), 7.61–7.53 (m, 3H), 7.03 (d, *J* = 16.4 Hz, 1H), 6.90 (d, *J* = 16.4 Hz, 1H), 6.76 (d, *J* = 10.1 Hz, 1H), 6.68 (s, 1H), 5.92 (d, *J* = 10.1 Hz, 1H), 3.86 (s, 3H), 1.52 (s, 6H), and 1.44 (s, 6H).

^13^C NMR (100 MHz, Acetone-*d*_6_) δ 175.88, 160.60, 155.53, 155.24, 153.49, 144.58, 131.86, 131.29, 130.79, 129.05, 126.23, 116.02, 113.90, 112.80, 112.49, 110.89, 108.00, 77.82, 70.17, 61.95, 29.74, and 27.49.

IR (thin film): 1735, 1716, 1471, 1434, 1420, 1396, 1315, 1023, 773, and 651 cm^−1^

m.p.: 497.4 °C

**22:**^1^H NMR (400 MHz, Chloroform-*d*) δ 7.96–7.85 (m, 2H), 7.58–7.40 (m,3H), 6.84–6.69 (m, 2H), 5.74 (d, *J* = 10.1 Hz, 1H), 5.02 (s, 1H), 4.88 (s, 1H), 4.42 (dd, *J* = 8.0, 5.2 Hz, 1H), 3.91 (s, 3H), 3.17 (m, 2H), 2.46 (s, 1H), 1.90 (s, 3H), 1.52 (s, 3H), and 1.50 (s, 3H).

^13^C NMR (100 MHz, Chloroform-*d*) δ 177.45, 161.22, 156.73, 156.17, 153.81, 147.53, 131.71, 131.44, 130.37, 129.08, 126.17, 116.34, 112.87, 112.29, 110.93, 110.64, 108.06, 78.23, 75.59, 62.87, 30.01, 28.55, 28.39, and 17.88.

IR (thin film): 1621, 1522, 1405, 1329, 1244, 1064, 829, 702, and 539 cm^−1^.

m.p.: 492.8 °C

To a solution of **27** (100 mg, 0.22 mmol) in anhydrous MeOH (5 mL) was added HCl (0.3 mL). The reaction mixture was stirred at room temperature for 0.1 h. EtOAc was added to the reaction mixture. The organic layer was extracted three times with saturated sodium bicarbonate solution (100 mL × 3) and the organic layers were washed with brine (30 mL × 3), dried over anhydrous Na_2_SO_4_, filtered, and concentrated in vacuo. The crude product was purified by flash chromatography (hexane:EtOAc = 20:1) to afford **28** (72.93 mg, 76% yield) as a yellow solid.

^1^H NMR (400 MHz, Chloroform-*d*) δ 7.94–7.82 (m, 2H), 7.59–7.47 (m, 3H), 7.41 (d, *J* = 16.5 Hz, 1H), 6.88 (d, *J* = 16.5 Hz, 1H), 6.64 (s, 1H), 6.54 (d, *J* = 10.1 Hz, 1H), 5.81 (d, *J* = 10.1 Hz, 1H), 5.17 (s, 1H), 5.14 (s, 1H), 2.48 (s, 3H), 2.08 (s, 3H), and 1.54 (s, 6H).

^13^C NMR (100 MHz, Chloroform-*d*) δ 177.10, 169.53, 161.81, 156.43, 155.29, 155.14, 142.85, 137.72, 131.73, 131.68, 131.50, 129.13, 126.25, 118.03, 117.18, 115.53, 112.72, 112.53, 111.02, 108.18, 78.47, 28.46, 21.12, and 18.18.

IR (thin film): 3589, 1991, 1770, 1735, 1540, 1472, 1457, 1420, 1362, 1288, 670, and 577cm^−1^

m.p.: 414.2 °C

To a solution of **28** (0.10 g, 0.23 mmol) in anhydrous THF (20 mL) was added 3M LiOH (2 mL). The reaction mixture was stirred at 60 °C for 3 h. EtOAc was added to the reaction mixture. The organic layers were washed with brine (30 mL × 3), dried over anhydrous Na_2_SO_4_, filtered, and concentrated in vacuo. The crude product was purified by flash chromatography (hexane:EtOAc = 15:1) to afford **2** (87.49 mg, 97% yield) as a yellow solid.

The ^1^H NMR and ^13^C NMR spectra of **2** are summarized in Appendix A.

Compound **3** and **1** was synthesized by following a similar procedure as that of **28.**

The ^1^H NMR and ^13^C NMR spectra of **3** and **1** are summarized in Appendix A.

Other experimental procedures and characterization data (^1^ H NMR,^13^ C NMR, and HRMS) can be found in the Appendix A.

## 4. Conclusions

In conclusion, we have accomplished the first total synthesis of lineaflavones A, C, and D and their analogues starting from commercially available raw materials. The key methods for the preparation of these compounds involve I_2_-catalyzed oxa-Michael addition, aldol reaction, Claisen rearrangement, and Schenck ene reaction. Besides this, we have developed five new routes to synthesize fifty-three natural product analogues, which provided the groundwork to explore structure–activity relationship studies.

## Data Availability

Data is contained within the article and supplementary material.

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
