# Peer review of "Total Synthesis of Lineaflavones A, C, D, and Analogues"

_molecules, 2023, doi:10.3390/molecules28052373_

Round 1

Reviewer 1 Report

Dear Authors,

You have described an impressive amount of work, with well detailed protocols. I have only a few suggestions.

You should elaborate (in the introduction) on what type of biological acivitie you are looking for.

Page 1, line 27: "...compounds with new biological..."

Page 1 line 42 remove "This"

Best Regards

Author Response

You have described an impressive amount of work, with well detailed protocols. I have only a few suggestions.

You should elaborate (in the introduction) on what type of biological acivitie you are looking for.

Response:Thanks for the advice. We have made the appropriate changes in the revised manuscript. (page 1, line 41)

Page 1, line 27: "...compounds with new biological..."

Response:Thanks for the advice. " compounds with biological " has been changed to " compounds with new biological " in the revised manuscript. (page 1, line 27)

Page 1 line 42 remove "This"

Response:Thanks for the advice. " This The isolated compounds " has been changed to " The isolated compounds " in the revised manuscript. (page 2, line 45)

Reviewer 2 Report

The submitted manuscript describes the considerable synthesis endeavour to prepare three recently isolated natural flavanoids, Lineaflavones A, C and D and a large number of unnatural analogues. The structures of these natural products have therefore been confirmed through synthesis for the first time.

The methods used for the preparation of the compounds are similar to those used for other, different, substituted flavones which have some, but not all of these substituents, and at different C-positions on the molecule.

The methodology is clearly explained and the outcomes of the synthetic routes are easily understandable. I was happy to see that the authors also report on conditions  that were unsuccessful, which is a help to others in their future compounds.  

A large number of analogues were prepared to exploit the methodology developed, including adding unnatural sidechains, cut-down analogues and rearranged examples. 

The experimental data is well reported and the supplementary data and NMR shows the compounds have been prepared to an acceptable purity. 

Overall, I recommend publication of this work with the following suggestions to improve the article

1. For the conversion of 21 to 1 (the methylation) the authors should provide details of the methylating agents used in these attempts. The bases were listed but were variety of methylating agents explored.

2. Could the authors hypothesise why the outcome occurred for the change in acids to give 1/3 in table 1 rather than just give the results?

3.  I would not agree with the sentence "The tertiary alcohol underwent methylation to give 1 as the major 118 product by using the condition of entry 4." The alcohol was most likely substituted for methanol so did not undergo methylation, methylation of 3 (via addition) was most likely the pathway.

4. The authors should add more detail as to why the choice of analogues were prepared, why were certain esters and ethers, for example, chosen and why those halogenated aromatics rather than any other substituent type? 

5. in the conclusion the authors state that they now have a library for SAR testing - they should state which assays/biological indications they consider the compounds could be suitable for.

Author Response

  1. For the conversion of 21 to 1 (the methylation) the authors should provide details of the methylating agents used in these attempts. The bases were listed but were variety of methylating agents explored.

Response:Thanks for the advice. We have made the appropriate changes. We used iodomethane and dimethyl sulfate as methylation agents, but we failed to detect the yield of compound 1. (page 4, line 111)

  1. Could the authors hypothesise why the outcome occurred for the change in acids to give 1/3 in table 1 rather than just give the results?

Response:Thanks for the advice. The tertiary carbon, with a carbon-carbon Double Bond at its α position, tend to undergo elimination reactions at high concentrations of acids. Lone-pair electrons on the oxygen atom of methanol and tertiary carbenium ion combine to generate substitution product. With the decreasing concentration of hydrogen ions in solution, the trend of hydrogen ions leaving of complex becomes more clearly.

  1. I would not agree with the sentence "The tertiary alcohol underwent methylation to give 1 as the major 118 product by using the condition of entry 4." The alcohol was most likely substituted for methanol so did not undergo methylation, methylation of 3 (via addition) was most likely the pathway.

Response:Thanks for the advice. We have made the appropriate changes in the revised manuscript. (page 4, line 123)

  1. The authors should add more detail as to why the choice of analogues were prepared, why were certain esters and ethers, for example, chosen and why those halogenated aromatics rather than any other substituent type? 

Response:Thanks for the advice. We explored five new routes to synthesize fifty-three flavonoid derivatives with novel structures. We introduced various substituents to increase structural diversity, which provided groundwork to explore structure-activity relationship studies.

  1. in the conclusion the authors state that they now have a library for SAR testing - they should state which assays/biological indications they consider the compounds could be suitable for.

Response:Thanks for the advice. Lineaflavones A, C and D are natural flavonoids isolated from the aerial parts of Tephrosialinearis by Spiteller and co-workers in 2020. The anti-inflammatory effects of the isolated compounds were evaluated by measuring the levels of IL-6 and TNF-α and the tested compounds inhibited the production of IL-6 and TNF-α. Therefore, newly synthesized derivatives could be suitable for evaluating anti-inflammatory activity.

Reviewer 3 Report

Presented article describes the first total synthesis of recently isolated natural flavonoids with anti-inflammatory activity - lineaflavones A, C, D. Moreover several series of their synthetic analogues were prepared providing the foundation for further massive biological activity research.

Unfortunately characteristics of new compounds are incomplete. There are no IR spectral data and melting points in Supplementary or Experimental Section. In addition overall  yields of lineaflavones A, C, D are not given anywhere in the text.

I accept the manuscript after minor revision.

Other minor remarks:

Line 27: The text contains a link to the article “Total Synthesis and Biological Evaluation of Siladenoserinol A and its Analogues” (Angewandte Chemie (International ed. in English) 57(18):5147-5150. DOI: 10.1002/anie.201801659). It is unclear how this article is related to the presented one.

Line 31: “Tephrosialinearis” The space is missing.

Line 42: “This The isolated …” This sentence is unclear. Check the English please (delete This).

Line 391: “Coclusions” Correct the spelling.

Schemes 4 and 5: The names of radicals are cyclobutyl (for compound 29d) and cyclohexyl (for compounds 29e and 40e).

References 12, 24and 28 contain full names instead initials.

Author Response

Presented article describes the first total synthesis of recently isolated natural flavonoids with anti-inflammatory activity - lineaflavones A, C, D. Moreover several series of their synthetic analogues were prepared providing the foundation for further massive biological activity research.

Unfortunately characteristics of new compounds are incomplete. There are no IR spectral data and melting points in Supplementary or Experimental Section. In addition overall yields of lineaflavones A, C, D are not given anywhere in the text.

Response:Thanks for the advice. IR spectral data and melting points of new compounds have been given in the revised Supporting Information. Overall yields of lineaflavones A, C, D have been given in the revised Supporting Information.

I accept the manuscript after minor revision.

Other minor remarks:

Line 27: The text contains a link to the article “Total Synthesis and Biological Evaluation of Siladenoserinol A and its Analogues” (Angewandte Chemie (International ed. in English) 57(18):5147-5150. DOI: 10.1002/anie.201801659). It is unclear how this article is related to the presented one.

Response:Thanks for the advice. We removed the link in the revised manuscript. (page 1, line 27)

Line 31: “Tephrosialinearis” The space is missing.

Response:Thanks for the advice. We have made the appropriate changes. (page 1, line 32)

Line 42: “This The isolated …” This sentence is unclear. Check the English please (delete This).

Response:Thanks for the advice. " This The isolated compounds " has been changed to " The isolated compounds " in the revised manuscript. (page 2, line 45)

Line 391: “Coclusions” Correct the spelling.

Response:Thanks for the advice. " Coclusion " has been changed to " Conclusion " in the revised manuscript. (page 11, line 402)

Schemes 4 and 5: The names of radicals are cyclobutyl (for compound 29d) and cyclohexyl (for compounds 29e and 40e).

Response:Thanks for the advice. We have made the appropriate changes in the revised manuscript. (page 5 and page 6)

References 12, 24and 28 contain full names instead initials.

Response:Thanks for the advice. We have made the appropriate changes in the revised manuscript. (page 12, line 445 and page 13, line 473, line 482)